

# Reviews on common objectives and evaluation indicators for risk communication activities from 2011 to 2017

Akiko Sato[1], Kaori Honda[1], Kyoko Ono[2], Reiko Kanda[3], Takehiko I. Hayashi[4], Yoshihito Takeda[5], Yoshitake Takebayashi[1], Tomoyuki Kobayashi[1,6] and Michio Murakami[1]

[1] Department of Health Risk Communication, Fukushima Medical University School of Medicine, Fukushima, Japan
[2] Research Institute of Science for Safety and Sustainability, National Institute of Advanced Industrial Science and Technology, Ibaraki, Japan
[3] Quantum Medical Science Directorate, National Institute for Quantum and Radiological Science and Technology, Chiba, Japan
[4] Center for Health and Environmental Risk Research, National Institute for Environmental Studies, Ibaraki, Japan
[5] Graduate School of Environment and Information Sciences, Yokohama National University, Kanagawa, Japan
[6] Research Fellow of Japan Society for the Promotion of Science, Tokyo, Japan

Corresponding author
Michio Murakami, michio@fmu.ac.jp

## ABSTRACT

**Background.** Risk communication is widely accepted as a significant factor for policy makers, academic researchers, and practitioners in diverse fields. However, there remains a lack of comprehensive knowledge about how risk communication is currently conducted across fields and about the way risk communication is evaluated.

**Methodology.** This study systematically searched for materials from three scholarly search engines and one journal with a single search term of "risk communication." The eligibility assessment selected peer-reviewed articles published in English that evaluated risk communication activities. Emphasis was placed on articles published in recent years accounting for about half of the pre-selected ones. Data on field of study, intervention timing, target audience, communication type, and objectives/evaluation indicators was extracted from the articles. Patterns of objectives/evaluation indicators used in risk communication activities were compared with those of the definitions and purposes of risk communication stated by relevant organizations. Association analysis was conducted based on study fields and objectives/evaluation indicators.

**Results.** The screening process yielded 292 articles that were published between 2011 and 2017 in various fields, such as medicine, food safety, chemical substances, and disasters/emergencies. The review process showed that many activities were performed in the medical field, during non-/pre-crisis periods. Recent activities primarily targeted citizens/Non-Profit Organizations (NPOs), and was disseminated in the form of large group or mass communication. While "knowledge increase," "change in risk perception and concern alleviation," and "decision making and behavior change" were commonly addressed in practice, "trust-building" and "reduction in psychological distress" were rarely focused. The analysis also indicated that the medical field tends to perform risk communication at the individual or small group level, in contrast to the

food safety field. Further, risk communications in the non-/pre-crisis period are more likely to aim at "changes in risk perception and concern alleviation" than those in the crisis period. Risk communications that aim at "changes in risk perception and concern alleviation" are likely to be presented in a large group or mass communication, whereas those that aim at "decision making and behavior change" are likely to be conducted at the individual or small group level.

**Conclusion**. An overview of recent activities may provide those who engage in risk communication with an opportunity to learn from practices in different fields or those conducted in different intervention timings. Devoting greater attention to trust building and reduction in psychological distress and exploring non-citizen/NPO stakeholders' needs would be beneficial across academic and professional disciplines.

# INTRODUCTION

The COVID-19 pandemic made the world—from politicians, scholars and practitioners to individual citizens—desperate for accurate, timely information. As of June 2020, there is no single day during which one does not hear updates or stories related to COVID-19. Not only reliable information, but myths and false messages have also spread rapidly, causing great confusion. Such incorrect information sometimes results in unnecessary fear or unrealistic hope among people (*World Health Organization, 2020a*; *Shiloh Vidon, 2020*). In response to the upsurge in demand from affected nations, the International Federation of Red Cross and Red Crescent Societies (IFRC), the United Nations Children's Fund (UNICEF), and the World Health Organization (WHO) jointly issued *the COVID-19 Global Response: Risk Communication and Community Engagement (RCCE) Strategy* (*International Federation of Red Cross and Red Crescent Societies, United Nations Children's Fund & World Health Organization, 2020*). The world is now keen on effective risk communication.

Although risk communication is now gaining tremendous attention, it is not new. In 1989, the United States National Research Council published *Improving Risk Communication* (*National Research Council, 1989*) and introduced an influential concept of risk communication by calling it:

An interactive process of exchange of information and opinion among individuals, groups, and institutions. It involves multiple messages about the nature of risk and other messages, not strictly about risk, that express concerns, opinions, or reactions to risk messages or to legal and institutional arrangements for risk management (*National Research Council, 1989*).

The concept has been ardently applied to avert the occurrence of different risks in life and reduce their impacts on human health, property and the environment (*Fischhoff, 1995*; *Covello & Sandman, 2001*; *International Risk Governance Center, 2017*). Apart from public health emergencies such as COVID-19, risk communication can take place in other areas, involving health care, food safety, and chemical substances (*Glik, 2007*; *Lopez-Gonzalez*

*et al., 2015*; *Tiozzo et al., 2011*; *MacDonald Gibson et al., 2013*). Risk communication is embedded in many aspects of everyday life.

In response to a growing awareness of the importance of risk communication, numerous organizations which are responsible for communicating about risk management strategies have presented their views on risk communication to facilitate its application. Table 1 summarizes examples of the definitions and purposes of risk communication stated by some leading organizations in their respective areas, involving United Nations organizations, the European Union, and other intergovernmental organizations, as well as national government departments and agencies. The table illustrates that organizations consider risk communication to be the transfer and exchange of risk-related information and opinions. Common and similar purposes of risk communication across the organizations are (1) knowledge increase, (2) communication satisfaction, (3) change in risk perception and concern alleviation, (4) reduction in psychological distress, (5) trust building, (6) decision making and behavior change, and (7) self-efficacy improvement. Some organizations also include ''facilitation of mutual understanding'' and ''citizen participation in policy making'' as a part of the purpose of risk communication.

The dynamic nature of risk communication, with these diverse purposes, makes it complex to transfer concepts into practice. As Lundgren and McMakin describe in *Risk Communication: A Handbook for Communicating Environmental, Safety, and Health Risks* (*Lundgren & McMakin, 2013*), risk communication may require different communication methods (e.g., oral, visual, other, or combined approaches; face-to-face and mass communication) and different concerned parties depending on risk characteristics, the surrounding circumstances, audience, and communication purposes. It is not necessarily straightforward to comprehend the purpose and method of risk communication or how its effects are measured in a given field. Further, it is even more cumbersome to grasp overall trends in risk communication practices and evaluation methods across different fields.

There are studies that have reviewed risk communication activities and their evaluation methods within a specific field (*Zipkin et al., 2014*; *Stewart, 1995*). However, the authors find that there is still a lack of synergetic research that provides a comprehensive overview. For that reason, this study attempted to identify and summarize the main objectives, approaches, and evaluation indicators applied for risk communication activities across fields. Furthermore, this study sought to investigate whether any discrepancies existed between prominent organizations' perspectives and expectations on risk communication and their respective practices.

## METHODOLOGY

### Inclusion and exclusion criteria

The inclusion criteria used to select empirical studies were (1) that the study evaluates risk communication activities in any field, and (2) that the study is written in English. This study also included previous research that did not directly evaluate risk communication but did ask implementers, such as medical professionals, about the objectives and effects of their activities relevant to risk communication. These studies were included because they

Sato et al. (2020), *PeerJ*, DOI 10.7717/peerj.9730

**Table 1** Definitions and purposes of risk communication as stated by select international and national organizations.

| Field | Organization | Definition | Purpose | Main purposes of risk communication[a] | | | | | | |
|---|---|---|---|---|---|---|---|---|---|---|
| | | | | 1 | 2 | 3 | 4 | 5 | 6 | 7 |
| Chemical substances | Organization for Economic Co-operation and Development (*Renn & Kastenholz, 2000*) (cited *Covello, von Winterfeldt & Slovic (1986)*) | The act of conveying or transmitting information between interested parties about (a) levels of health or environmental risks; (b) the significance or meaning of health or environmental risks; or (c) decisions, actions, or policies aimed at managing or controlling health or environmental risks. | | ✓ | | | | | | |
| | *European Chemicals Agency (2010)* | | Helping to build trust among organizations that risks are being adequately assessed and managed; assisting with making better decisions on how to address risks; helping to ensure smoother implementation of risk management policies; helping to empower and reassure the general public; helping to bridge the gap between real risks and perceived risks ; and helping to prevent crises from developing and managing them when they do occur. | ✓ | | ✓ | | ✓ | ✓ | |
| | *United States Environmental Protection Agency (2019)* | The process of informing people about potential hazards to their person, property, or community. | To help residents of affected communities understand the processes of risk assessment and management, to form scientifically valid perceptions of the likely hazards, and to participate in making decisions about how risk should be managed. | ✓ | | ✓ | | | | |
| | Ministry of the Environment, Japan[b] (*Ministry of the Environment Japan, 2002*) (cited the Chemical Society of Japan (*Chemical Society of Japan, 2001*)) | Sharing accurate information and exchanging opinions between citizens, industry, government, and other interested parties on health and environmental risks related to chemical substances. | To increase awareness and understanding of the relevant risk and its management and to build a trust relationship among all concerned stakeholders, and reduce the risk through demanding and providing information and exchanging opinions between stakeholders . | ✓ | | ✓ | | ✓ | | |

**Table 1** (*continued*)

| Field | Organization | Definition | Purpose | Main purposes of risk communication[a] | | | | | | |
|---|---|---|---|---|---|---|---|---|---|---|
| | | | | 1 | 2 | 3 | 4 | 5 | 6 | 7 |
| Food safety | Food and Agriculture Organization of the United Nations and World Health Organization (*Food and Agriculture Organization of the United Nations & World Health Organization, 1998*; *Food and Agriculture Organization of the United Nations & World Health Organization, 2016*) | The exchange of information and opinions concerning risk and risk-related factors among risk assessors, risk managers, consumers and other interested parties. | To enable people to protect their health from food safety risks by providing information that enables them to make informed food safety decisions , to facilitate dialogue and understanding among all interested stakeholders, and to improve the overall effectiveness of the risk analysis process. | | | | | | | |
| | Codex Alimentarius Commission (*Codex Alimentarius Commission, 2018*) | The interactive exchange of information and opinions throughout the risk analysis process concerning risk, risk-related factors and risk perceptions, among risk assessors, risk managers, consumers, industry, the academic community and other interested parties, including the explanation of risk assessment findings and the basis of risk management decisions. | Risk communication should: (i) promote awareness and understanding of the specific issues under consideration during the risk analysis; (ii) promote consistency and transparency in formulating risk management options/recommendations; (iii) provide a sound basis for understanding the risk management decisions proposed; (iv) improve the overall effectiveness and efficiency of the risk analysis; (v) strengthen the working relationships among participants; (vi) foster public understanding of the process, so as to enhance trust and confidence in the safety of the food supply; (vii) promote the appropriate involvement of all interested parties ; and (viii) exchange information in relation to the concerns of interested parties about the risks associated with food. | ✓ | | ✓ | | ✓ | ✓ | |
| | European Food Safety Authority (*European Food Safety Authority, 2017*) | | To assist stakeholders, consumers and the general public to understand the rationale behind risk-based decisions and, to help them make balanced judgements about the risks that they face in their own lives. | | | | | | | |

**Table 1** (*continued*)

| Field | Organization | Definition | Purpose | Main purposes of risk communication[a] | | | | | | |
|---|---|---|---|---|---|---|---|---|---|---|
| | | | | 1 | 2 | 3 | 4 | 5 | 6 | 7 |
| | | | Effective risk communication can contribute to the success of a risk management program by: (1) ensuring that consumers are aware of the risks associated with a product and thereby use or consume it safely; (2) building public confidence in risk assessment and management decisions and the associated risk/benefit considerations; (3) contributing to the public's understanding of the nature of a risk or risks; and (4) providing fair, accurate, and appropriate information , so that consumers are able to choose among a variety of options that can meet their own "risk acceptance" criteria. | ✓ | | ✓ | | ✓ | ✓ | |
| Food safety & medicine | United States Food and Drug Administration (*United States Department of Health and Human Services & Food and Drug Administration, 2011*; *United States Department of Health and Human Services & Food and Drug Administration, 2012*) | | Share information, change beliefs, change behavior. | | | | | | | |
| | | Risk communication activities fall into two broad categories: (1) interactively sharing risk and benefit information to enable people to make informed judgments about use of FDA-regulated products and (2) providing guidance to relevant industries about how they can most effectively communicate the risks and benefits of regulated products. | (Examples listed as intermediate outcomes that can lead to the improvement of overall public health are as follows:) (1) improved understanding of the risks and benefits of regulated products by the multiple audiences with whom FDA communicates, including relevant international audiences; (2) increased public awareness of crisis events and the increased likelihood that affected individuals or groups will take recommended actions; (3) increased public satisfaction with FDA as an expert and credible source of information about regulated products; and (4) increased confidence that target audiences are getting useful, timely information as it becomes available, to help them make informed choices. | ✓ | ✓ | ✓ | | ✓ | ✓ | |
| Medicine & disasters | World Health Organization (*Gamhewage, 2014*; *World Health Organization, 2019*) | The two-way and multi-directional communications and engagement with affected populations. | To share information vital for saving life, protecting health and minimizing harm to self and others; to change beliefs ; and/or to change behavior . | | | | | | | |
**Table 1** (*continued*)

| Field | Organization | Definition | Purpose | Main purposes of risk communication[a] | | | | | | |
|---|---|---|---|---|---|---|---|---|---|---|
| | | | | 1 | 2 | 3 | 4 | 5 | 6 | 7 |
| | | The exchange of real-time information, advice and opinions between experts and people facing threats to their health, economic or social well-being. | To enable people at risk to take informed decisions to protect themselves and their loved ones. | ✓ | | ✓ | | | ✓ | |
| | United States Nuclear Regulatory Commission (*Persensky et al., 2004*) | An interactive process used in talking or writing about topics that cause concern about health, safety, security, or the environment. | (Examples listed:) (1) providing information to the public about numerous issues, including inspection findings and their significance, changes to regulatory requirements, security and safeguards issues, or how the decision-making process works; (2) to learn about stakeholder concerns, perceptions about risks, expectations about involvement in risk management decisions, or local information that will assist in risk analysis; (3) building/restoring trust and relationships; (4) to ask stakeholders for input in a decision-making process; and (5) influencing people's behavior and perceptions about risk. | ✓ | | ✓ | | ✓ | ✓ | |
| | *United States Department of Health and Human Services & Centers for Disease Control and Prevention (2018)* | Risk communication provides the community with information about the specific type (good or bad) and magnitude (strong or weak) of an outcome from an exposure or behavior. Typically, risk communication is a discussion of a negative outcome and the probability that the outcomes will occur. | Risk communication can be employed to help an individual make a choice about a behavior such as smoking, getting vaccinated, or undergoing a medical treatment. | ✓ | | | | | ✓ | |

**Notes.**

Underlined parts correspond to indicators identified in this study.

[a] 1 = knowledge increase, 2 = communication satisfaction, 3 = change in risk perception and concern alleviation, 4 = reduction in psychological distress, 5 = trust building, 6 = decision making and behavior change, 7 = self-efficacy improvement.

[b] Translated by an author of this article (AS).

provided insight into the objectives/evaluation of risk communication that is expected. Review studies, commentaries, conference proceedings, and books were excluded. Since abstracts were reviewed during the first round of eligibility assessment, materials that did not provide an abstract were excluded. Articles that discuss the procedures of future risk communication activities—meaning that the activities had not been implemented at the time of publication—were also excluded.

## Search strategy

Potential materials for this study were identified on April 18, 2018, through relevant search engines, namely, PubMed, ScienceDirect, and PsycINFO. These sources are accepted as the world's leading scholarly search engines since they provide access to peer-reviewed literature in a wide range of academic disciplines. The only search term used was "risk communication." The *Journal of Risk Research* was also included as a source for the material collection because this journal contains many study articles on the topic of risk communication that are not covered by these search engines. Initially, the material search was not limited by the year of publication to learn how the number of publications related to risk communication activities has shifted over time. After eliminating duplicates and articles that did not provide an abstract, and obtaining a solid idea about the number of possibly eligible studies, the authors used this information to determine the span of publication years from which studies would be included. This time span includes approximately half of the relevant materials published in recent years.

## Eligibility assessment

This study conducted two rounds of eligibility assessment. As it is briefly stated above, the first round was a review of only titles and abstracts of articles that were identified through the search engines and the *Journal of Risk Research*. This initial assessment was to pre-select materials from which to derive the eligibility criteria for this study and to obtain a broad picture of recent risk communication activities in order to finalize a plan for subsequent analysis. The second round involved a review of full texts of pre-selected articles to confirm their eligibility.

For the first round of eligibility assessment, the team established groups consisting of two researchers. Each researcher independently assessed assigned articles and determined whether the study (1) evaluated risk communication activities quantitatively, (2) assessed the objectives and/or effects of risk communication activities qualitatively, and (3) discussed the objectives and/or effects of risk communication activities based on prior experiences and/or existing scientific knowledge. For the second round, researchers were re-grouped, and a pair of researchers independently read assigned articles that they had not checked during the first round to confirm the eligibility of the articles and finalize the material selection.

The principal investigator of the research project (MM) coordinated this evaluation and selection process. MM checked all articles and developed a basic protocol for the eligibility assessment. In general, if both reviewers who checked a particular article agreed in their evaluation of its eligibility, the decision was accepted. When there was a disagreement, MM

facilitated discussions between the researchers to achieve consensus. When necessary, MM reflected the points of agreement in the protocols to ensure consistency in evaluation.

## Coding process

After completion of the first round of eligibility assessment, the research team discussed what data should be extracted and how the information should be labeled and coded. The team made decisions based on characteristics of risk communication activities learned from the first round of eligibility assessment, the international and national organizations' statements on risk communication (Table 1) and other relevant literature, as well as individual researchers' experience and expertise.

Researchers remained in the same group formed for the second round of eligibility assessment, and separately extracted data from each assigned article and coded it as follows:

- Evaluation approach: (1) quantitative, (2) qualitative, and (3) based on prior experience and/or existing scientific knowledge (see the criteria in "Eligibility Assessment").
- Study field: (1) medicine, such as health and pharmaceutical realms, (2) food safety, (3) chemical substances (other than food safety matters), (4) nuclear and radiological disasters/emergencies, (5) other disasters/emergencies, (6) climate change, and (7) other.
- Timing when a risk communication intervention was implemented in line with the phases in the disaster management cycle: (1) non-crisis or pre-crisis, including non-specified, (2) crisis, and (3) post-crisis, including recovery phase.
- Target audience: (1) citizens (e.g., individual citizens, residents, unspecified persons, and citizen groups) or Non-Profit Organizations (NPOs), and (2) other (e.g., government, professionals, and companies).
- Communication type: (1) individual/small group communication (e.g., doctor–patient–family communication and family communication), and (2) large-group/mass communication.
- Objective/indicator: (1) knowledge increase, (2) communication satisfaction, (3) change in risk perception and concern alleviation, (4) reduction in psychological distress, (5) trust building, (6) decision making and behavior change (e.g., risk acceptance, risk avoidance, and risk management, such as avoidance of unhealthy foods, seeking healthcare, disaster mitigation and preparedness, and community partnerships; attitude toward behavior and behavioral intention were also included in this category), (7) self-efficacy improvement, and (8) other.

With regard to "intervention timing", this study employed the three-stage approach proposed by *Coombs (2012)*. The term "crisis" in this study refers to the definition proposed by the same scholar (*Coombs, 2007*) as, "a significant threat to operations that can have negative consequences if not handled properly." The pre-crisis period involves the detection of warning signs relating to such crisis and prevention and/or preparedness. The crisis period concentrates on identifying the onset of a crisis, controlling the situation, and minimizing negative impacts. The post-crisis period concerns rehabilitation and full recovery from the crisis, evaluation of crisis management, and better preparation for future

crisis (*Coombs, 2012*). With regard to "objective/evaluation indicator", the researchers jointly determined how indicators should be classified by referring to the definitions and purposes of risk communication stated by the selected international and national organizations (Table 1).

Where applicable, multiple response categories were selected. If both reviewers who checked a particular article classified it the same way, the decision was accepted. When the two researchers differed, discrepancies were evaluated by a third researcher. When needed, the issues were discussed with MM until all concerned researchers reached an agreement on the article's classification.

Examples of evaluation indicators were drawn for this paper. Specifically, one example for each indicator was taken from the field of medicine, and another was from other fields due to the generally large number of relevant medicine-related articles. Examples were chosen based on the frequency of citation assessed on May 10, 2019 through Google Scholar and the clarity of applied methods. Even if some frequently cited studies targeted multiple indicators, they were referred to for only one indicator among all the applicable indicators.

### Statistical analysis

Data was entered into a Microsoft Office Excel spreadsheet. Excel was used to compute descriptive information on the collected data. Additionally, sets of Pearson's chi-squared test with Yates's continuity correction and Fisher's exact test were conducted to examine the associations by study field and by objective/evaluation indicator. The statistical analyses were performed with studies that belonged to a single category of all the variables except for the variable of "objective/evaluation indicator". "Nuclear and radiological disasters/emergencies," "climate change," and "other" from the study field variable were excluded because of their small size. For the same reason, the "crisis" group and "post-crisis" group were combined in the analysis on the associations by study field, and the "crisis" group was excluded in the analysis on the associations by objectives/evaluation indicators. For analyses involving study field, post hoc test (*Aoki, 2010*) was conducted to determine where differences occurred if an initial analysis identified a significant difference between study field and other variables. R (*R Development Core Team, 2020*) was used for the statistical analyses. Test results were considered significant at $P < 0.05$. $P$-value adjustment by Holm's method was applied for multiple comparisons.

## RESULTS

### Search results

Figure 1 summarizes the flow of the material search and selection for this study. The database search found 3,710 articles, mostly from PubMed (57%). Of those, 1,433 articles went through the first round, and 412 moved on to the second round of eligibility assessment. In the end, 292 articles published between 2011 and 2017 remained for review.

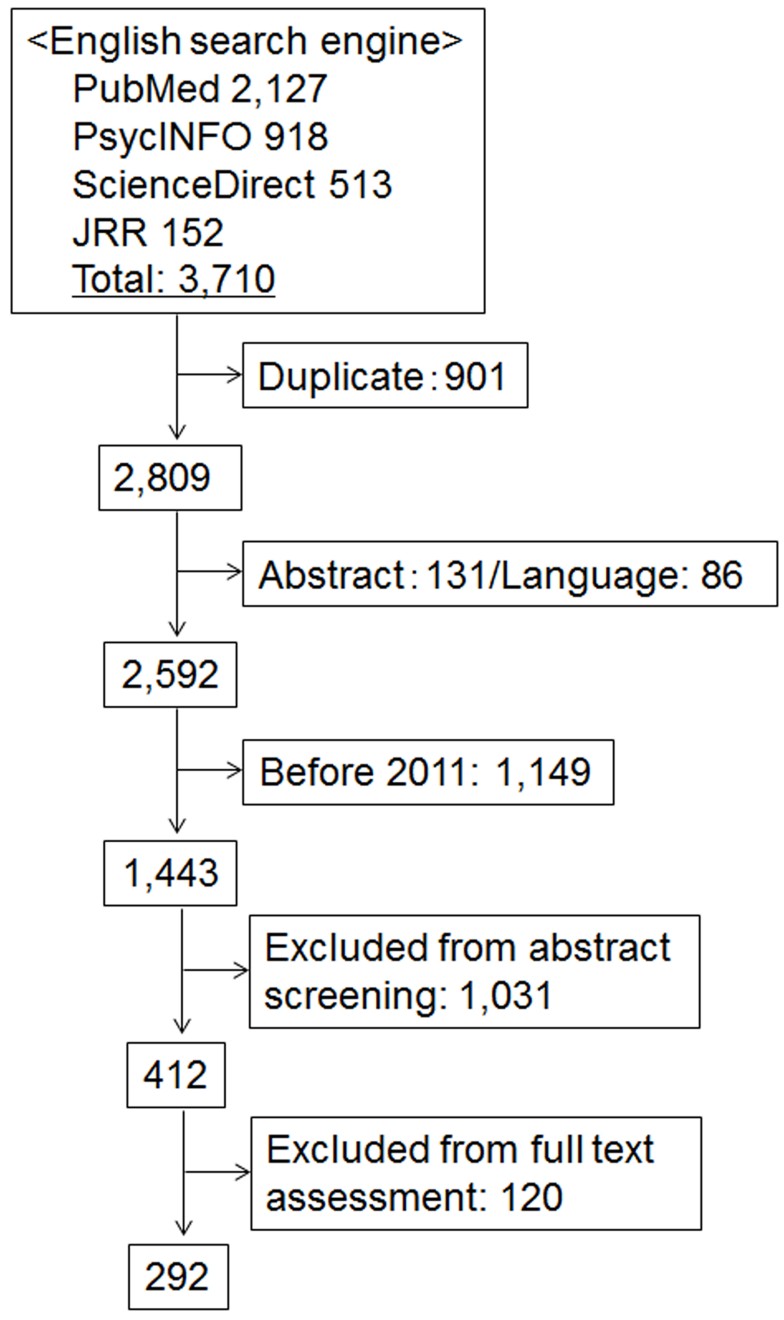

**Figure 1** **Flow diagram of material selection.** JRR = Journal of Risk Research.

## Characteristics of risk communication studies

The data generated for this study is provided in Table S1. The table contains basic information from all 292 articles. Figure 2 shows a descriptive summary of the data.

More than 80% of the studies quantitatively evaluated own risk communication practices. Over 60% were related to medicine. Studies classified as "other" included those addressing human–wildlife conflicts (*Lu et al., 2018*) and traffic safety (*Feenstra, Ruiter & Kok, 2014*;

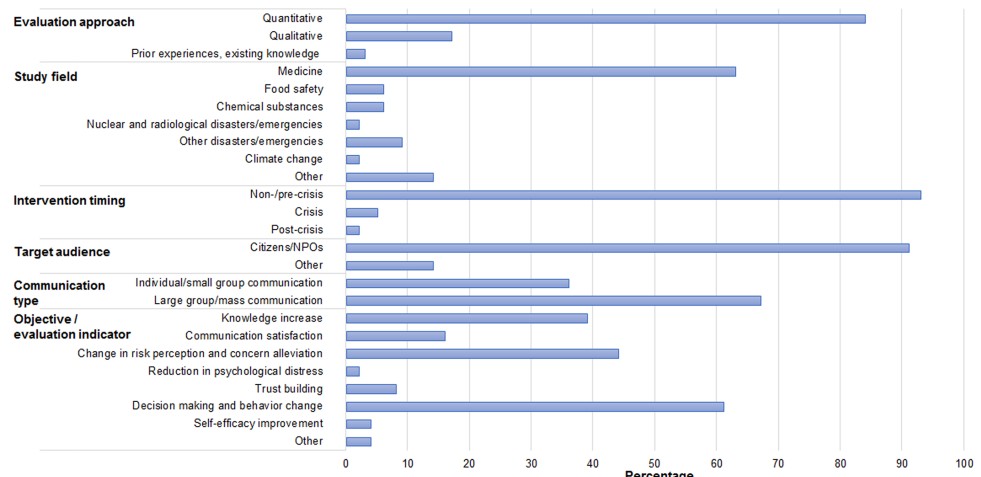

**Figure 2** Descriptive statistics of the study variables ($N = 292$).

*Wu & Weseley, 2013*), as well as studies that used a risk scenario or involved multiple risk domains to investigate effective means or to assess intrapersonal and other factors of risk communication (*Dawson, Johnson & Luke, 2017*; *Poortvliet & Lokhorst, 2016*). Five studies (2%) fell in multiple study fields. The vast majority of activities were implemented during a non-/pre-crisis phase (93%), and one study involved multiple phases. Over 90% of the studies targeted citizens/NPOs. Of those, 15 studies or 5% also approached other target groups, such as medical professionals and farmers. As for communication type, more than 60% were communications to a large group audience or the public. Of those, seven studies or 2% were also conducted in the form of individual/small group communication. Frequently-used objectives/evaluation indicators were "decision making and behavior change," "change in risk perception and concern alleviation," and "knowledge increase" (61%, 44%, and 39%, respectively). Examples of objectives/evaluation indicators are shown in Table 2.

## Comparison between risk communication definitions and purposes and main objectives/evaluation indicators

The authors of this study identified "knowledge increase," "change in risk perception and concern alleviation," and "decision making and behavior change" as areas of focus in all fields; these objectives are also discussed in the definitions and purposes of most organizations and agencies (Fig. 2, Table 1). Here, "knowledge" is about the risks of concern and related risk management policies and actions. "Change in risk perception" primarily focused on guiding individuals' subjective judgment of risk to align with available scientific evidence. Table 1 also illustrates that "reduction in psychological distress" does not generally appear in the selected organizations' definitions and purposes of risk communication, which is consistent with this study; this topic was rarely addressed in the studies reviewed in this research (2%).

Peer J

**Table 2  Evaluation examples from risk communication studies.**

| Indicator | Author(s), year of publication | Study field | Study description | Example(s) |
|---|---|---|---|---|
| Knowledge increase | *Brown et al. (2011)* | Medicine | This study assessed the relationships between health literacy, numeracy, and the ability to interpret graphs. Participants were asked to interpret different types of graphs in the context of breast cancer risk and make hypothetical treatment decisions. | Interpreting the risk of a new breast cancer occurring in the other breast following preventive surgical options based on the hypothetical information from the provided graphs, making a surgical option, and describing differences in remaining risk between surgical options. |
| | *Moussaïd, Brighton & Gaissmaier (2015)* | Chemical substances | This study analyzed social transmission of risk information by examining how messages on the risk of a controversial antibacterial agent changed when being passed from one person to another in a chain of up to 10 persons. | Information diversions and defects occurred while being transferred from one person to the next. |
| Communication satisfaction | *Garcia-Retamero & Cokely (2011)* | Medicine | This study evaluated the effectiveness of gain- and loss-framed messages and visual aids about sexually transmitted diseases (STDs) on participants' reactions to intervention material and their STD-related risk perception, attitude, behavioral intention, and behaviors. | Participants' evaluation on how interesting, involving, and informative the intervention material was. |
| | *Tiozzo et al. (2011)* | Food safety | This study evaluated the effectiveness of a campaign on salmonellosis on public risk awareness and knowledge on risk and prevention behavior. | Participants' evaluation of the usefulness of the campaign material. |
| Change in risk perception and concern alleviation | *Nan et al. (2015)* | Medicine | This study investigated the impact of evidence-oriented messages and narrative-type messages about human papillomavirus (HPV) on recipients' risk perception and vaccination intentions. | Participants' perceived susceptibility to HPV. |
| | *Binder et al. (2011)* | Other disasters/ emergencies | This study analyzed the influence of interpersonal discussions on residents' perceptions about the risks and benefits of the planned US National Bio- and Agro-Defense Facility. | Residents' perceived risk of negative impacts associated with the facility on their safety, health, and the environment. |
| Reduction in psychological distress | *Henneman et al. (2013)* | Medicine | This study assessed the effects of the provision of graphs in addition to frequency information about breast cancer on at-risk women's risk understanding, psychological wellbeing, and intention to have breast screening. | Psychological wellbeing measured by an adapted version of the Lerman Cancer Worry Scale (CWS) and the Dutch version of the six-item version of the state scale of the Spielberger State–Trait Anxiety Inventory. |
| | *MacDonald Gibson et al. (2013)* | Chemical substances | This study explored how probabilistic information influences risk understanding, opinions regarding risk/site management, risk perception, and concerns of residents who live nearby a closed site contaminated with unexploded ordnance. | Negative emotional reactions to the provided information: "How (worried, afraid, anxious) would you be about (getting hurt if you worked at the site, letting children play near the site, living near the site)?" |
| Trust building | *Besser, Anderson & Weinman (2012)* | Medicine | This study conducted interviews with patients with osteoporosis and collected their drawings to assess their views on the illness and treatment, as well as their conditions. | Doctor–patient relationship was reported as one motivation to adhere to medication regimen. |
| | *Cronin, Midgley & Jackson (2014)* | Other (genetic engineering) | This study introduced "Issues Mapping" to facilitate dialogues between different stakeholders, clarify different perspectives, and promote mutual understanding. It applied the techniques to social conflicts relating to genetic engineering issues. | Perceptions of genetic engineering including participants' trust in other stakeholders and their views on current debate in society. |
**Table 2** (*continued*)

| Indicator | Author(s), year of publication | Study field | Study description | Example(s) |
|---|---|---|---|---|
| Decision making and behavior change | *Lopez-Gonzalez et al. (2015)* | Medicine | This was an intervention study to see if communicating to people about cardiovascular diseases (CVDs) by using risk assessment tools (Framingham REGICOR and Heart Age) would lead to improvement in their CVD risk factors. | Changes in physical activity (number of sessions of physical activity per week), smoking behavior, and other modifiable risk factors, involving anthropometrical and blood pressure data. |
| | *Rabinovich & Morton (2012)* | Climate change | This study assessed the effect of people's beliefs about nature and science on their perspective about uncertainty in relation to climate change. | Participants' willingness to carry out positive environmental behaviors (e.g., reducing water use) and agree on a household carbon budget. |
| Self-efficacy improvement | *Harris, Sutherl & Hutchinson (2013)* | Medicine | This study analyzed the influence of parents' marital status, and parent–child sexual communication and relationship on male adolescents' knowledge regarding HIV and STDs, and their intentions and their implementation of preventive behaviors. | Six-item Condom Use Self-Efficacy scale (e.g., "I am confident that I know how to use a condom.") |
| | *Feenstra, Ruiter & Kok (2014)* | Other (traffic safety) | This study assessed the impacts of a school-based road safety program on risk perception, attitude, intention, and behaviors in relation to risky cycling among 9th–11th-grade students. | Perceived self-efficacy for safe cycling (e.g., controlling the bicycle and applying traffic rules) in comparison with peers. |

In contrast, "trust building" is frequently included among the proposed purposes of risk communication, whereas it was rarely addressed in the studies assessed in this research (8%). Further, while the organizations extended the target of risk communication in their statements to non-citizen parties, such as industries and media, those groups were rarely targeted in risk communication activities in the studies assessed in this research. Consequently, relevant indicators were not discussed in the study articles.

### Differences by study field

Table 3 shows the frequency data, which is cross-tabulated with study field and other variables. In most fields of study, risk communication activities were conducted in a non-/pre-crisis phase, whereas nearly half of the risk communications in the field of nuclear and radiological disasters/emergencies were conducted in a non-/pre-crisis phase, and the remaining were in a post-crisis phase. The table shows that the majority of risk communication targets citizens/NPOs. At the same time, 29% in the field of chemical substances targeted other groups. In the medical field, half of the risk communications were conducted at an individual level or in a small group, whereas risk communication in other fields was conducted mainly in a larger group or to an entire population of interest.

Table S2 shows the results of the analyses between study field and other variables. The series of analyses found significant associations between study field and communication type ($P < 0.01$). Multiple comparisons suggested a significant difference pertaining to communication type between risk communications in the field of medicine from those in food safety ($P < 0.05$).

### Differences by objective/evaluation indicator

Table 4 highlights the diversity in objectives and indicators of risk communication activities. For instance, the fields of food safety and other (i.e., non-nuclear/radiological) disasters and emergencies had a higher percentage in terms of risk communications aiming or addressing "trust building" (22% and 19%, respectively) than other fields, especially compared to the field of medicine (5%). The chemical substance field had a higher percentage (76%), and the field of nuclear and radiological disasters/emergencies had a lower percentage (33%) compared with other fields with regard to risk communications focusing on "decision making and behavior change."

Table 4 also suggests some percentage differences in intervention timing, target audience, and communication type by objective/evaluation indicator. The main objectives and indicators of risk communications conducted in a non-/pre-crisis period were "knowledge increase," "change in risk perception and concern alleviation," and "decision making and behavior change" (39%, 45%, and 62%, respectively). "Decision making and behavior change" was a main indicator for risk communications conducted in a crisis period (57%). "Knowledge increase," and "change in risk perception and concern alleviation" were the main indicators for risk communications conducted in a post-crisis period (83%, 50%, respectively). There was over 15% difference in "change in risk perception and concern alleviation" between risk communications targeting citizens/NPOs compared to

**Table 3  Intervention timing, target audience, and communication type by study field ($N = 292$).**

| | | Study field | | | | | | |
|---|---|---|---|---|---|---|---|---|
| | | Medicine ($n = 184$) | Food safety ($n = 18$) | Chemical substances ($n = 17$) | Nuclear and radiological disasters/ emergencies ($n = 6$) | Other disasters/ emergencies ($n = 27$) | Climate change ($n = 5$) | Other ($n = 40$) |
| | | $n$ (%) | $n$ (%) | $n$ (%) | $n$ (%) | $n$ (%) | $n$ (%) | $n$ (%) |
| Intervention timing | Non-/pre-crisis | 174 (95) | 16 (89) | 16 (94) | 3 (50) | 22 (81) | 5 (100) | 40 (100) |
| | Crisis | 9 (5) | 1 (6) | 1 (6) | 0 (0) | 4 (15) | 0 (0) | 0 (0) |
| | Post-crisis | 1 (1) | 2 (11) | 0 (0) | 3 (50) | 1 (4) | 0 (0) | 0 (0) |
| Target audience | Citizens/NPOs | 164 (89) | 18 (100) | 14 (82) | 5 (83) | 27 (100) | 5 (100) | 39 (98) |
| | Other | 31 (17) | 0 (0) | 5 (29) | 1 (17) | 0 (0) | 0 (0) | 3 (8) |
| Communication type | Individual/small group communication | 91 (49) | 0 (0) | 4 (24) | 1 (17) | 6 (22) | 0 (0) | 2 (5) |
| | Large group/mass communication | 96 (52) | 18 (100) | 15 (88) | 5 (83) | 23 (85) | 5 (100) | 38 (95) |

**Notes.**

Although the total number of studies included in the analysis was 292, the total number of each variable varies owing to the allowance of multiple responses. The percentages were based on the total number of each study field.

Sato et al. (2020), *PeerJ*, DOI 10.7717/peerj.9730

**Table 4  Study field, intervention timing, target audience, and communication type by objective/evaluation indicator ($N = 292$).**

| | | Objective/evaluation indicator | | | | | | | |
|---|---|---|---|---|---|---|---|---|---|
| | | Knowledge increase | Communication satisfaction | Change in risk perception and concern alleviation | Reduction in psychological distress | Trust building | Decision making and behavior change | Self-efficacy improvement | Other |
| | | n (%) | n (%) | n (%) | n (%) | n (%) | n (%) | n (%) | n (%) |
| Study field | Medicine ($n = 184$) | 78 (42) | 37 (20) | 67 (36) | 6 (3) | 10 (5) | 115 (63) | 10 (5) | 8 (4) |
| | Food safety ($n = 18$) | 7 (39) | 5 (28) | 10 (56) | 0 (0) | 4 (22) | 8 (44) | 0 (0) | 1 (6) |
| | Chemical substances ($n = 17$) | 9 (53) | 1 (6) | 7 (41) | 1 (6) | 0 (0) | 13 (76) | 0 (0) | 0 (0) |
| | Nuclear and radiological disasters/emergencies ($n = 6$) | 3 (50) | 1 (17) | 3 (50) | 0 (0) | 1 (17) | 2 (33) | 0 (0) | 0 (0) |
| | Other disasters/emergencies ($n = 27$) | 7 (26) | 1 (4) | 16 (59) | 0 (0) | 5 (19) | 16 (59) | 1 (4) | 2 (7) |
| | Climate change ($n = 5$) | 1 (20) | 0 (0) | 3 (60) | 0 (0) | 0 (0) | 2 (40) | 0 (0) | 0 (0) |
| | Other ($n = 40$) | 13 (33) | 2 (5) | 25 (63) | 0 (0) | 3 (8) | 22 (55) | 2 (5) | 2 (5) |
| Intervention timing | Non-/pre-crisis ($n = 273$) | 107 (39) | 44 (16) | 124 (45) | 7 (3) | 19 (7) | 168 (62) | 13 (5) | 12 (4) |
| | Crisis ($n = 14$) | 3 (21) | 1 (7) | 2 (14) | 0 (0) | 3 (21) | 8 (57) | 0 (0) | 1 (7) |
| | Post-crisis ($n = 6$) | 5 (83) | 2 (33) | 3 (50) | 0 (0) | 1 (17) | 2 (33) | 0 (0) | 0 (0) |
| Target audience | Citizens/NPOs ($n = 267$) | 106 (40) | 46 (17) | 119 (45) | 6 (2) | 20 (7) | 163 (61) | 13 (5) | 12 (4) |
| | Other ($n = 40$) | 15 (38) | 5 (13) | 10 (25) | 2 (5) | 3 (8) | 24 (60) | 1 (3) | 2 (5) |
| Communication type | Individual/small group communication ($n = 104$) | 42 (40) | 19 (18) | 33 (32) | 5 (5) | 5 (5) | 74 (71) | 5 (5) | 6 (6) |
| | Large group/mass communication ($n = 195$) | 75 (38) | 28 (14) | 96 (49) | 2 (1) | 17 (9) | 110 (56) | 9 (5) | 7 (4) |
| OVERALL | ($N = 292$) | 115 (39) | 47 (16) | 128 (44) | 7 (2) | 22 (8) | 177 (61) | 13 (4) | 13 (4) |

**Notes.**

The total number of each variable varies because of the allowance of multiple responses. Percentages were based on the total number of each value.

risk communications targeting other groups (45% vs. 25%). The same was observed for "change in risk perception and concern alleviation" and "decision making and behavior change" between risk communications conducted at an individual/small group level and risk communications conducted in larger groups (32% vs. 49% and 71% vs. 56%, respectively).

The analyses (summarized in Table S3) revealed a significant association between study field and "trust building" ($P < 0.05$), between intervention timing and "change in risk perception and concern alleviation" ($P < 0.05$), between communication type and "change in risk perception and concern alleviation" ($P < 0.05$), and between communication type and "decision making and behavior change" ($P < 0.05$). Multiple comparisons did not find any significant difference between study fields in terms of "trust building."

## DISCUSSION

This study was implemented to obtain a comprehensive picture of recent risk communication practices across academic fields. Many established practices were taking place in the medical field, during a non-/pre-crisis period, targeting citizens/NPOs, and in the form of large group or mass communication. There are multiple possible explanations for the findings. First, the medical industry has a generally larger number of publications (*Piro, Aksnes & Rørstad, 2013*), and some medical issues, such as chronic diseases, are generally common or well-known problems. Second, there are naturally many more individuals who are at risk but not yet affected by certain threats (i.e., in the "non-/pre-crisis period"), compared with those who are already affected. Furthermore, communication that takes place during or shortly after the occurrence of an unexpected event is often called "crisis communication," as opposed to "risk communication" (*United States Department of Health and Human Services & Centers for Disease Control and Prevention, 2018*). Finally, risk communication is a central tool employed to protect public health and safety (*International Risk Governance Center, 2017*; *Glik, 2007*). It is understandable that the ultimate beneficiaries became the target population of many risk communication activities.

The communality in the use of increased knowledge, a change in risk perception, and behavior change as objectives and evaluation indicators of recent activities can be explained by the Health Belief Model (*Hochbaum, 1958*; *Rosenstock, 1974a*; *Rosenstock, 1974b*) which was developed by social psychologists and is one of the most influential theoretical models. This model proposes that these three primary objectives are closely linked and that knowledge influences individual perceptions about risks and can guide people to perform recommended preventive behavior, which leads to better health outcomes. The United States Food and Drug Administration also lists these three as central objectives of risk communication (*United States Department of Health and Human Services, Food and Drug Administration, 2011*). The usefulness of addressing these domains is empirically supported by the active application of this cognitive and behavioral model in various public health settings (*Sharifirad et al., 2009*; *Tola et al., 2016*; *Ghaffari et al., 2012*).

Risk communication activities are taking diverse approaches correlating with varying objectives as discussed in the literature (*National Research Council, 1989*; *Covello &*

*Sandman, 2001*; *International Risk Governance Center, 2017*; *Lundgren & McMakin, 2013*); yet, this study suggests that there are some repeated patterns in the implementation. Risk communication activities that aim at changing risk perceptions of target audience and/or alleviating their concerns are likely to be conducted in the form of large group or mass communication. In contrast, risk communication activities that aim at supporting decision making and behavior change are likely to be conducted in the form of an individual/small group communication. This pattern corresponds with the finding of *Edwards et al. (2000)*. They reviewed literature on risk communication in health care and found one-to-one communication to be highly effective in decision making and behavioral change because such communication is suitable to address individuals' circumstances, and their specific needs and concerns.

Some important gaps in current practices were revealed in this study. For example, the majority of the target audience of risk communication activities was identified as citizens or Non-Profit Organizations (NPOs). However, stakeholders concerning risk communication also include academics and professionals, governments, media, industries, individual producers, and emergency-response agencies (*Codex Alimentarius Commission, 2018*; *United States Department of Health and Human Services & Centers for Disease Control and Prevention, 2018*). It is critical that all stakeholders receive regular risk communication training to impart up-to-date knowledge about relevant risks and to foster and maintain their ability to engage in managing risks in a collaborative manner. The marginalization of the need to focus on reducing psychological distress and trust building in risk communication activities is another concern. The importance of cultivating trust relationships and addressing mental health components is well understood. However, past studies have pointed out challenges including differences between scientists and laypersons in technical knowledge, risk perceptions and access to information, as well as complex and severe social problems surrounding people who are at risk of or affected by crisis, such as poverty, displacement and loss of livelihoods (*Fischhoff, 1995*; *Covello, von Winterfeldt & Slovic, 1986*; *Renn, Webler & Johnson, 1991*; *World Health Organization, 2020b*). This study indicates that further efforts are needed to address these challenges.

This study has several limitations. The chosen methodology for material search may have limited the investigation from discovering additional relevant materials, although this study used large search engines that cover a broad range of topics and the widely accepted term of "risk communication." In addition, relevant conference presentations, books and non-English materials were not included for review. The methodological appropriateness and validity of individual studies were not evaluated during the material selection process. There may have been studies whose quality of evidence was suboptimal. Lastly, because this study focused only on risk communication activities conducted in recent years, it did not evaluate in this paper how they have changed over time, which limited the scope of the analysis.

In spite of the above limitations, this study makes a significant contribution to risk communication research and provides some practical insights. For instance, the authors found that "changes in risk perception and concern alleviation" are less focused upon during times of crisis compared to other times. Additionally, "reduction in psychological

distress" has been neglected or not prioritized in risk communication activities. Yet, a large-scale and prolonged crisis like the COVID-19 pandemic has demonstrated that long-lasting social isolation, disruption of daily lives, and an uncertain end to a crisis will cause tremendous stress and exhaustion. Some people may begin to feel ill or depressed, while other people may grow weary of being extremely cautious (*Rogers et al., 2020*; *United States Department of Health and Human Services & Centers for Disease Control and Prevention, 2020*; *Brewer, 2020*). Addressing psychological impacts such as these is vital and should be part of the emergency response.

Suggested future work includes the investigation of the effectiveness of risk communication activities across academic fields. It is critical to determine whether effectiveness differs based on risk characteristics and risk communication objectives and approaches. To address the gaps in recent practices, key lessons can be drawn from risk communication activities aimed at trust building and reduction in psychological distress, as well as from risk communication targeting non-citizen/NPO stakeholders. Future research can also explore how a crisis evolves and determine its implications for risk communication activities. For instance, it is useful to assess changes in the approach toward communication about COVID-19 along with the emergence and spread of the disease, in consideration of its profound and diverse impacts on society and individuals.

## CONCLUSION

While risk communication has been implemented in a variety of ways for diverse objectives, this study revealed some overall trends in the objectives, approaches and evaluation indicators applied for recent risk communication activities. At the same time, the results of analysis also suggest that there are some patterns in implementation; associations exist between the study field and the communication type, and between the objectives/evaluation indicators and the intervention timing and communication type. These facts may provide useful insights to those who are involved in risk communication in designing and evaluating their activities. This study also identified the limited attention in current practices to cultivating trust building and reduction in psychological distress, as well as targeting non-citizen/NPO groups. Addressing these gaps is an important way forward for a sustainable path toward effective risk management and better resilience.

## ACKNOWLEDGEMENTS

This article was prepared after the authors added analyses, results, and discussion to a report for the *Research on the Health Effects of Radiation* organized by the Ministry of the Environment, Japan (http://www.env.go.jp/chemi/chemi/rhm/R0104e_3.pdf). We thank our colleagues who are in charge of other parts of the larger research project that encompasses this study. They provided valuable input during the preparation and implementation of this study. The views discussed in this paper are those of the authors and do not necessarily reflect the positions of those colleagues, the affiliated institutions, or the funding agency.

### Funding
This study was supported through the Research Project on Health Effects of Radiation organized by the Ministry of the Environment, Japan. The funders had no role in study design, data collection and analysis, decision to publish, or preparation of the manuscript.

### Grant Disclosures
The following grant information was disclosed by the authors:
The Research Project on Health Effects of Radiation organized by the Ministry of the Environment, Japan.

### Competing Interests
The authors declare there are no competing interests.

### Author Contributions
- Akiko Sato performed the experiments, analyzed the data, prepared figures and/or tables, authored or reviewed drafts of the paper, and approved the final draft.
- Kaori Honda, Kyoko Ono, Reiko Kanda, Takehiko I. Hayashi, Yoshihito Takeda, Yoshitake Takebayashi and Tomoyuki Kobayashi performed the experiments, analyzed the data, authored or reviewed drafts of the paper, and approved the final draft.
- Michio Murakami conceived and designed the experiments, performed the experiments, analyzed the data, prepared figures and/or tables, authored or reviewed drafts of the paper, and approved the final draft.

### Data Availability
The raw data are available as a Supplemental File.

### Supplemental Information
Supplemental information for this article can be found online at http://dx.doi.org/10.7717/peerj.9730#supplemental-information.

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
