# Peer review of "Reviews on common objectives and evaluation indicators for risk communication activities from 2011 to 2017"

_PeerJ, doi:10.7717/peerj.9730_

## Round 0.1 · original submission · Major Revisions

The manuscript provides what seems to be a scoping review of Risk Communication which is an important topic. There is a need for greater focus and more brevity to reduce the length of the different sections and allow readers to focus on what is important. Some tables should be changed to figures to allow easier visualization of differences across fields since the actual values are not of main concern but rather the pattern. Figure 2 is not needed. The part where there is comparison with definitions from organizations should not be part of the Methods and Results. In general, it is important to ensure that the information is mentioned where it is required and not in other sections. Please check my detailed comments in the attached pdf.

·

Basic reporting

I appreciate the opportunity to review this study. Risk communication is both an important and timely topic. I believe this manuscript has a good potential for publication, provided that revisions are made, primarily those contained in other sections of this review.

The manuscript does a reasonably good job of introducing the topic of risk communication, although in general I find the manuscript lacking in relevant examples to engage the reader. The quality of the writing is good overall (some minor suggestions are made in the “additional comments” section). It does not appear that similar work has been done recently, making a review paper appropriate.

The paper’s abstract and introduction places considerable emphasis on evaluation of risk communication strategies, but does not appear to delve deeply into the results of these evaluations – that is to say, any assessment of which risk communications strategies that the reviewed papers found to be more or less effective. In my opinion this is a significant oversight, as at present the paper only describes some evaluation methods and notes their frequency, but does not discuss their effectiveness.

Experimental design

The study design in this manuscript generally appears sound. However, it is difficult to evaluate many of the author’s choices because they are presented only as fact (for example, in noting that the years 2011 to 2017 were chosen for evaluation) without providing insights on why these decisions were made. As examples:

1. Is there some important event that took place in or before 2011 that made that year to be the early cutoff? Or was this simply a resource issue?

2. What is the significance of the particular academic search engines used for this study? Given that PubMed (a comprehensive database, but focused exclusively on medical research) was one of the search engines, it is no surprise that most of the articles were medical in nature. Was any consideration given to other academic search engines?

One of the main findings of the manuscript is that the lack of academic research appearing through these particular search parameters in many fields means that little risk communication is happening in those fields (see for example, lines 608-610). It does not appear that the authors have explored alternative reasons why those fields did not show up in their search. For example, could the databases possibly not contain those fields of study? Is it possible that those fields of study use a term other than “risk communication” for this activity? Or some other reason?

Validity of the findings

This study seems to lack substantive conclusions. At present, the conclusions section is mostly a series of generalities that could easily have been included in the introduction section. Only a few sentences (around 600-610) are truly informed by the work. I’d recommend that the authors find ways to better express the results of the study within the conclusions section, regardless of what those conclusions are.

The authors also have not clearly explained why key decisions were made (see “study design”) which at present is a substantial impediment to evaluating the manuscript.

Additional comments

General suggestion: are there any ways that the authors could create more graphical information for this manuscript? It is currently table-heavy (understandably given the subject matter) and some additional informative figures could make the article more engaging for readers.

Line 27: Risk communication is used both in “academic fields” as well as through various types of practitioners, which is acknowledged in the paper but not in the abstract.

Line 31: The use of “search engines” in the abstract implies the use of popular search engines (i.e. Google or Yahoo) as opposed to academic or scholarly search engines.

Line 79-90: Given the mention of H7N9, it would seem prudent to provide any relevant insights related to COVID-19 in the revisions. Being an all-encompassing topic for much of the world, any relevant linkages would be beneficial to bring forward.

Lines 84-87: This is a very long run on sentence. I recommend splitting this into several different thoughts, which will be easier for the reader to understand.

Lines 114-116: Although this is a succinct statement of what was done, the reasons for the various choices are not clear. For example, is there some significance to the year 2011? Why these particular search engines?

Lines 146-154: This is an extremely long and complex sentence which is very confusing. Recommend splitting this into several different thoughts.

Lines 164-166: It is not clear how the codes were chosen. Given that many of the codes are fairly general, it’s unclear how bundled or unbundled the codes were meant to be, and how they were developed. This is key context for the reader.

Lines 277-278: Is this referring to “Likert Scales” or “Likert-type data”? Scales are made from aggregating more than one Likert-type data point. The terms appear to be mixed here.

Line 424: The abstract and introduction place considerable emphasis on the evaluation of risk communication methods, but here it implies this was only a development of “methods by which it is practiced.”

Reviewer 2 ·

Basic reporting

Clear and unambiguous, professional English used throughout. The article must be written in English and must use clear, unambiguous, technically correct text. The article must conform to professional standards of courtesy and expression.
The paper is well written and does not need any further improvement of the language. Sentences are easy to read and enable me to fully understand what the authors intended.

Literature references, sufficient field background/context provided. The article should include sufficient introduction and background to demonstrate how the work fits into the broader field of knowledge. Relevant prior literature should be appropriately referenced.
In the Introduction section, the authors address the definition of “risk communication” and its relevance for difference research and policy fields. Therefore, it is a challenging task to give an exhaustive definition in order to represent this variety (lines 62-63). The authors made a good job but I would suggest to add some other references, e.g. Lundgren & McMakin (2009) and, more generally, the work done by professionals, consultants, … who tired to translated definitions into practices. Through the years, risk communication was born as a discipline, to be defined and addressed, and followed by research and tentative interventions to make it works and be effective in understanding and properly managing risks, no matter the field. So, in this sense, I would stress that risk communication definition and aims are also given by this kind of literature. The authors will refer to some examples of these sources further ahead.

Professional article structure, figures, tables. Raw data shared. The structure of the article should conform to an acceptable format of ‘standard sections’ (see our Instructions for Authors for our suggested format). Significant departures in structure should be made only if they significantly improve clarity or conform to a discipline-specific custom.
The article structure is well organised and it fits with the suggested format for PEERJ Review articles.

Figures should be relevant to the content of the article, of sufficient resolution, and appropriately described and labeled.
Ok

All appropriate raw data have been made available in accordance with our Data Sharing policy.
Yes

Is the review of broad and cross-disciplinary interest and within the scope of the journal?
Yes, it is confirmed both by the search strategy and the results themselves.

Has the field been reviewed recently? If so, is there a good reason for this review (different point of view, accessible to a different audience, etc.)?
To my knowledge and brief online research, no recent review addressed this field. Recent works preferably reviewed risk communication in specific field, e.g. food safety, cancer, human health in general.
Does the Introduction adequately introduce the subject and make it clear who the audience is/what the motivation is?
Yes, see above for other comments to improve this section, even if it is already well organised.

Experimental design

Article content is within the Aims and Scope of the journal.
In my opinion, this paper falls within this following point:
PeerJ does not publish in the Physical Sciences,the Mathematical Sciences, the Social Sciences, or the Humanities (except where articles in those areas have clear applicability to the core areas of Biological, Environmental, Medical or Health sciences).
Rigorous investigation performed to a high technical & ethical standard. The investigation must have been conducted rigorously and to a high technical standard. The research must have been conducted in conformity with the prevailing ethical standards in the field.
Confirmed
Methods described with sufficient detail & information to replicate. Methods should be described with sufficient information to be reproducible by another investigator.
Overall, the reader understands the process made by the authors to search for the relevant articles and define the final simple. However, some questions about the Survey Methodology section arose and need major revision at some points. I would also express my concern about the choice to insert in the search procedures articles published in Japanese. How did this focus on the Japanese language serve and improve your analysis and results? I don’t want to prefer a language out of another one, I just want to say that including papers from Japanese literature could possibly bias e.g. against the fields of intervention, for example Japanese literature could be more focused on nuclear risk communication, whereas the Dutch one could be more focussed on flood risk communication. Your aim was to conduct an international review, so many other languages could have been considered and added to the search and results, according to pre-defined criteria of inclusions.
- Line 116: why did you choose the 2011-2017 period? I can understand 2017, as you have begun the analysis in 2018, but I wonder why you went back to 2011 precisely
- Line 117-118: I would not have included this kind of work, as they did not evaluate interventions. How many articles fall into this category? Please consider not including them in the study
- Please identify better the beginning and the end of the description of both the first and the second rounds of eligibility assessment. E.g. the first round goes from line 126 to 157 and refers only to title and abstract, then the second…
- Line 140: there is no need of the statement in brackets
- Lines 143: they were divided into six groups: why? Which were these groups? What were their purpose?
- Lines 143-146: why and how did you decide these three criteria? Just because of the study aims, other? Why did you assign the retrieved articles to one of these tree categories (as described in lines 142-154) already in the initial screening process? Would it had been enough just to assign a YES/NO to the articles according to the fact that they evaluated risk communication? Basically, I don’t understand why you performed this classification twice (lines 166-167), as in my opinion it was unnecessary in the initial phase. In addition, what about participatory approach in evaluating risk communication practices? Did you find anything about?
- Line 172, in line with the phases in the disaster management cycle: please provide a reference for that
- Lines 178-180: non-profit organisations: I would re-allocate this group, because these figures cannot be considered as “citizens”. In addition, it is reductive to put together in the same category “other” figures that have different roles in risk communication and intervene at different levels. Please, consider revising this classification and definition of target groups. Accordingly, I see the -same problem in lines 181-182, I refer to “(2) other (e.g. mass communication)”, where “other” could have been substituted by specific communication types and formats. As I can see from Table 1, “Other communication type” (N=214) doubles “Individual/small group communication”, making me supposing that it is not a residual category.
- Lines 166-189: in these lines you report all categories assigned to the selected articles: which coding approach did you adopt to code? How and when did you identify, define and name the categories?
- Lines 183-189: this is a great concern for me, as what you named “indicators used for evaluation, including desired or intended impacts on target audience” are not to be intended as indicators, because they clearly represent intended impacts of risk communication interventions. I mean, categories from 1 to 8 are example of aims to be achieved by e.g. a risk communication campaign, and they are not indicators to be used to measure whether an aim has been reached or not (to this extent, you could for example use the expression “evaluation instruments” used in line 268). You already address this difference in lines 263-267. Please, fix this point. Consider revising in Table 1 too and all concerned sections (e.g. lines 272-275, 315-356, …).
- Lines 213-217: why did you searched these documents for definitions? You had better use these sources to identify a set of evaluation criteria upon which to base and guide your analysis, for instance to fix the point I mentioned above. Finally, consider the presence of these documents and guidance in your Introduction.
Is the Survey Methodology consistent with a comprehensive, unbiased coverage of the subject? If not, what is missing?
The subject under investigation – i.e. risk communication has been extensively searched in the main academic search engines.
Are sources adequately cited? Quoted or paraphrased as appropriate?
Yes
Is the review organized logically into coherent paragraphs/subsections?
Yes

Validity of the findings

Impact and novelty not assessed. Negative/inconclusive results accepted. Meaningful replication encouraged where rationale & benefit to literature is clearly stated.
- Lines 251-252: why did you mentioned nuclear/radiological disaster and climate change here? I would expect stressing these points in the discussion section, together with an explanation of why there is a need to highlight this result
- Lines 277-290: I am sorry but I did not catch the meaning of these lines and their usefulness in this part of the manuscript
- Table 2: did you consider the possibility to indicate the research techniques used to evaluate risk communication? The “Instrument” column could be filled in with the technique, e.g. online survey, focus group, usability study, … the methodological approach (qualitative, quantitative, both…) and the theoretical framework beneath.
- Lines 320-323: I am not sure that this is as higher as intended, for example in this study field the aim “change in risk perception” has a higher percentage.
- Lines 358-385: this part would be better moved to the Introduction section, because it does not show results. Moreover, going ahead with the reading, and I am more and more persuaded that the documents you refer to in these lines can be considered as those who guided your selection of the criteria for the analysis. In this sense, this could shed more light to your results, in terms of what coincide and differ, as you already mentioned (e.g. lines 400-402).
- Lines 414-420: I take your point in pointing out this difference among definitions but this sentence seems to be out of the initial premises of the study, as it is not devoted to compare risk communication definition. So please consider where and how to collocate this part.
Conclusions are well stated, linked to original research question & limited to supporting results. The conclusions should be appropriately stated, should be connected to the original question investigated, and should be limited to those supported by the results. In particular, claims of a causative relationship should be supported by a well-controlled experimental intervention. Correlation is not causation. Speculation is welcome, but should be identified as such.

Is there a well developed and supported argument that meets the goals set out in the Introduction?
- Lines 425-427: where did you discuss these results? This could be an additional frame to classify and discuss results.
Does the Conclusion identify unresolved questions / gaps / future directions?
Overall, the Discussion section tries to interpret results in order to explain the differences in risk communication research and practices that emerged from the retrieved and analysed papers. The authors offer interesting interpretations and comments, that I definitely agree with, but that one could have expected. Some suggestions for future directions are given in the Conclusion but I try to ask you whether more intriguing insights could have been gained to better guide for example future research on risk communication, both in terms of applicability and evaluation? Any other suggestions with particular reference to crisis situations? Crisis situation is one of the three main categories risk communication is divided into, and what’s happening today due to the new SARS-Cov-2 pandemic shows that risk communication needs to be re-thought and this can be made only after a thorough evaluation of interventions made during the emergency.

Additional comments

Please make any general comments not covered by the 3 areas above. This text is sent to the author.
In this study, the authors tried to measure how much risk communication has been done and how it was evaluated, whenever evaluated, spacing throughout different fields of investigation and application of this discipline. To this extent, the authors reviewed recent English and Japanese literature to (lines 104-105)
- Select papers within these premises
- Appraise evaluation processes
- Identify indicators used to evaluate risk communication activities across sectors.
I really appreciated the possibility to read and review this work, as I fully agree that there is need to re-think what risk communication is and the extent to which it is able to help people facing risks. One of the major challenges of risk communication, as it is within this study’s aims and as the new pandemic is showing us, is that risk communication practices lack effectiveness, especially during crisis and emergency situation. All principles defined and applied so far seem to fail and this utterly calls for a deeper understanding of what is risk communication and what mines its credibility and applicability. In this sense, lines 106-108 stand out as guidance for future work to fix this point.

---

## Round 0.2 · accepted · Accept

The paper is greatly improved. Congratulations!

·

Basic reporting

I appreciate the opportunity to review this study for a second time. The authors have addressed my concerns in this section.

Experimental design

The authors have addressed my concerns around study design, and I do not have any new concerns for them to address.

Validity of the findings

The authors have addressed my concerns that impact the validity of the findings.

Additional comments

I do not have any additional comments for the authors at this time.

Reviewer 2 ·

Basic reporting

No comments, these sections have been definitively improved according to the reviewers' suggestions.

Experimental design

the method section is more and more clear and easy to follow; data collection has been better organised into coherent paragraphs.

Validity of the findings

Results have been more deeply discussed in terms of uncovering what risk communication research and practice need to effectively help risk governance both in peace and crisis times.

Additional comments

I would suggest to changing the title into:
Evaluating risk communication interventions: a review of research and practices across fields